# Non-Equilibrium Thermodynamics View on Kinetics of Autocatalytic Reactions—Two Illustrative Examples

**DOI:** 10.3390/molecules26030585

**Published:** 2021-01-22

**Authors:** Miloslav Pekař

**Affiliations:** Faculty of Chemistry, Brno University of Technology, Purkyňova 118, 612 00 Brno, Czech Republic; pekar@fch.vut.cz; Tel.: +420-5-4114-9330

**Keywords:** autocatalysis, non-equilibrium thermodynamics, rate equation, reaction kinetics

## Abstract

Autocatalytic reactions are in certain contrast with the linear algebra of reaction stoichiometry, on which rate equations respecting the permanence of atoms are constructed. These mathematical models of chemical reactions are called conservative. Using a non-equilibrium thermodynamics-based theory of chemical kinetics, it is shown how to introduce autocatalytic step into such (conservative) rate equation properly. Further, rate equations based on chemical potentials or affinities are derived, and conditions for the consistency of rate equations with the entropic inequality (the second law of thermodynamics) are illustrated. The theory illustrated here can be viewed as a tool for verifying and generalizing traditional mass-action kinetics by means of modern non-equilibrium thermodynamics, which is able to deal also with such rather problematic cases.

## 1. Introduction

“A reaction whose rate is proportional to the concentration of a product is said to be autocatalytic.” is written in the comprehensive textbook on physical chemistry written by Berry et al., on p. 879, [1]. Similar definitions of autocatalysis and autocatalytic reactions can be found in other textbooks (e.g., [2,3]) or papers ([4,5]). Houston [3] broadens this definition on reacting systems with multiple reactions: “...a product of one reaction appears as a catalyst in the same or another coupled reaction.” The same author, as well as, for example, Schuster [6], give a basic example of the autocatalytic reaction in the form A + X → 2X; its rate (in the forward direction) is usually given by the product of rate constants and concentrations: $kc_{A}c_{X}$. Autocatalytic reactions are known and studied for more than a century [4]. They are known for their complex behavior like self-organization or oscillatory phenomena [6] and play or are believed to play important roles in the system’s chemistry [7], the natural selection [6], the emergence of life [4], or prebiotic evolution [5].

Kinetics of autocatalytic reactions is studied by both deterministic and stochastic approaches [6], often using formal chemical reactions like Lotka’s scheme [3] A + X → 2X, X + Y → 2Y, Y → Z, which are non-stoichiometric and do not follow the permanence of atoms (mass conservation); such models are based on the atom-free stoichiometry [8].

During the last years, we have been developing a thermodynamics-based theory of chemical kinetics, i.e., a theory naturally consistent with thermodynamics. This theory originates in the non-equilibrium continuum thermodynamics of the linear fluids, which represents many reacting systems of interest in chemistry. The theory respects the permanence of atoms (the atomic structure of the reacting components) and belongs to conservative approaches in the terminology used by Érdi and Tóth [8]. The thermodynamics and the basics of its application in kinetics are presented in the book [9]. More details on the related theory of kinetics can be found in papers [10,11]. Very briefly, the thermodynamics shows that in the case of linear fluids, the reaction rate is a function of temperature and concentrations only. This is just a general statement, giving no explicit form for this function. Therefore, this function is approximated by a polynomial of a suitable degree in concentrations with temperature-dependent coefficients. The polynomial is simplified by its application on an equilibrium where the reaction rate vanishes by definition and where expressions for equilibrium constants can be used. The resulting polynomial, the final form of rate equation, is called the thermodynamic polynomial [9,10]. The whole procedure can be viewed as verification and generalization of the traditional kinetic mass-action law by non-equilibrium thermodynamics.

The methodology of the thermodynamic polynomial operates on independent reactions only, which is followed as a linear-algebraic result of the permanence of atoms (mass conservation) in chemical reactions; in other words, as a result of reaction stoichiometry [12]. It is thus in contrast with the formal—atom-free stoichiometry—approach mentioned above. This raises a question on the application of the stoichiometry-based methods on autocatalytic reactions, which, in principle, as illustrated below, are in contrast with the linear algebra of stoichiometry and can be viewed as a “purely chemical” concept. To answer such a question, this note analyzes two simple examples of autocatalytic reactions using the referenced methodology and shows that and how it simply introduces autocatalytic step into rate equation (thermodynamic polynomial).

## 2. Results and Discussion

### 2.1. Single Reaction

Perhaps the simplest autocatalytic reaction ([6], see Introduction) expressed generally in terms of atom-conserving components (that is not in just general symbols of substances) is written as
(R1)X+BX=2X+B

(Because the discussed thermodynamic approach is a mathematical theory, which views stoichiometric equations as equations, the “=” symbol is used instead of the double arrow common in kinetics). The autocatalysis is seen in the fact that two reactant X molecules are produced, while one of them is consumed in the forward direction of (R1). Traditionally, the rates in the forward and reversed directions would be written in this way:(1)$$\vec{r}=\vec{k}c_{X}c_{\mathrm{BX}};\;\;\overset{↼}{r}=\overset{\leftarrow }{k}c_{X}^{2}c_{B}$$

The thermodynamic methodology starts with writing down the compositional matrix and finding its rank (for details, see [9], pp. 150–151, [12]). Numbering the “atoms” as 1=X, 2=B and components as 1=X, 2=BX, 3=B, the matrix has the dimension 2×3 and is
(2)$$\left\|S\right\|=\left[\begin{array}{ccc} 1 & 1 & 0\\ 0 & 1 & 1 \end{array}\right]$$

Its rank h=2. The total number of the components in this reacting mixture is n=3; thus, the number of the independent reactions is n−h=1 [9] (p. 153), [12]. The stoichiometric matrix ‖P‖ of the only one independent reaction is of the dimension (n−h)×n=1×3 and should fulfill the condition ${\left\|P\right\|\left\|S\right\|}^{T}=\left\|0\right\|$ [13]. The entry $\left\|P^{p\alpha }\right\|$ is the stoichiometric coefficient of α in the reaction *p*. If the matrix ‖P‖ is written generally as $\left[\begin{array}{ccc} a & b & c \end{array}\right],$ the condition results in:(3)a+b=0,  b+c=0.

The reaction R1 calls for two stoichiometric coefficients of X that are forbidden by (3), and this reaction cannot be considered and selected as an independent reaction in the discussed reacting mixture. Further, from (3), it is followed that a=−b, i.e., the components X and BX should be stated at the opposite sides of the stoichiometric equation (of independent reaction). Nevertheless, the thermodynamic methodology still enables to formulate rate equation with the autocatalytic step.

Suitable and allowable selection of the independent reaction is
(R2)BX= B+X.

Its equilibrium constant is (ideal system, unit standard concentration)
(4)$$K=\frac{c_{B,\mathrm{eq}}c_{X,\mathrm{eq}}}{c_{\mathrm{BX},\mathrm{eq}}}.$$

The rate of this reaction (J) as a function of temperature and concentrations, as stated in the Introduction, is first approximated by a polynomial of the third degree in concentrations, which is sufficient to obtain meaningful results (the first or second degree does not lead to the occurrence of an autocatalytic term in the resulting rate equation). The simplification procedure gives the following thermodynamic polynomial (for details, see Appendix A):(5)$$\begin{array}{c} J=k_{010}\left(c_{\mathrm{BX}}-K^{-1}c_{X}c_{B}\right)+k_{020}\left(c_{\mathrm{BX}}^{2}-K^{-1}c_{X}c_{\mathrm{BX}}c_{B}\right)+k_{110}\left(c_{X}c_{\mathrm{BX}}-K^{-1}c_{X}^{2}c_{B}\right)+k_{011}\left(c_{\mathrm{BX}}c_{B}-K^{-1}c_{X}c_{B}^{2}\right)\\ +k_{120}\left(c_{X}c_{\mathrm{BX}}^{2}-K^{-1}c_{X}^{2}c_{\mathrm{BX}}\right) \end{array}$$

(The subscripts of the polynomial coefficients reflect the powers of the concentrations in the corresponding term, e.g., $k_{110}c_{1}c_{2}\equiv k_{110}c_{X}c_{\mathrm{BX}}$; see also ref. [14,15]).

As suggested earlier [10,11], the individual terms in the thermodynamic polynomial are interpreted in the view of the traditional mass-action kinetics as representing steps in the reaction scheme hidden in the polynomial. The scheme corresponding to (5) is:(R3a)BX= X+B,
(R3b)2BX=X+ BX+B,
(R3c)X+BX=2X+B,
(R3d)BX+B=X+2B,
(R3e)X+2BX=2X+ BX.

The selected independent reaction (R2) is recovered as (R3a), the supposed autocatalytic step (R1) as (R3c). Although this autocatalytic step is excluded by the starting linear algebra of stoichiometry, cf. discussion below (3), the methodology of the thermodynamic polynomial allows its presence in the rate equation. Regardless of the occurrence of the single independent reaction, its reaction rate (5) contains up to four additional reaction steps, forming the 5-step scheme (R3). Among them, there are additional autocatalytic steps like (R3d) or (R3e). The full scheme (R3) should be viewed as a mathematical result, i.e., a mathematically allowable set of steps in kinetic equations. Some or even many of these steps could not be those that real-chemistry should state or detect, i.e., which steps are reliable and which remain as “pure mathematics”. This is illustrated after Equation (6); for more detailed discussion, see ref. [14].

From the linear algebra of the stoichiometry of the studied reaction mixture (for details, see [9], p. 154, [12]), it is followed that
(6)$$J^{X}=J=J^{B},\;\;\;J^{\mathrm{BX}}=-J$$
where $J^{i}$ is the component rate. When only (R1) really occurs, the mass-action rate equation is obtained $k_{110}\neq 0,k_{010}=k_{020}=k_{011}=k_{120}=0$. If both (R1) and (R2) occur, then $k_{110}\neq 0,k_{010}\neq 0,k_{020}=k_{011}=k_{120}=0$.

The thermodynamic methodology presented in this paper includes also proper transformations of rate equation into the function of chemical potentials or affinities. We have illustrated both ways in the following simplified thermodynamic polynomial:(7)$$J=k_{010}\left(c_{\mathrm{BX}}-K^{-1}c_{X}c_{B}\right)+k_{110}\left(c_{X}c_{\mathrm{BX}}-K^{-1}c_{X}^{2}c_{B}\right).$$

The transformation to the function of chemical potentials is straightforward and is based on the relationship between the equilibrium constant and standard chemical potentials:(8)$$-RT\ln K=\sum_{\alpha }\mu_{\alpha }^{o}P^{1\alpha }=\mu_{X}^{o}-\mu_{\mathrm{BX}}^{o}+\mu_{B}^{o}.$$

The traditional model of chemical potential as a function of concentration (in ideal systems):(9)$$\mu_{\alpha }=\mu_{\alpha }^{o}+RT\ln\left(\frac{c_{\alpha }}{c^{o}}\right)$$
where unit standard concentration will be used. The result is
(10)$$J=\left(k_{010}\exp\frac{-\mu_{\mathrm{BX}}^{o}}{RT}\exp\frac{\mu_{\mathrm{BX}}}{RT}+k_{110}\exp\frac{-\mu_{X}^{o}-\mu_{\mathrm{BX}}^{o}}{RT}\exp\frac{\mu_{X}+\mu_{\mathrm{BX}}}{RT}\right)\left(1-\exp\frac{\mu_{X}-\mu_{\mathrm{BX}}+\mu_{B}}{RT}\right).$$

The transformation to the function of affinities requires the knowledge of the relationships between chemical potentials and affinities. Note that in the proper mathematical derivation, not only the traditional chemical affinity but also another affinity, called the constitutional affinity, appears [13]. There is only one chemical affinity (of the only one independent reaction) in our system defined as (cf. [9], p. 181):(11)$$A=\sum_{\alpha }\mu_{\alpha }P^{1\alpha }=\mu_{X}-\mu_{\mathrm{BX}}+\mu_{B}$$

There are two constitutional affinities ($B^{\sigma }$) defined by (cf. [9], p. 182)
(12)$$B^{\sigma }=\overset{3}{\underset{\alpha =1}{\sum }}\overset{2}{\underset{\tau =1}{\sum }}\mu_{\alpha }S_{\tau \alpha }f^{\sigma \tau };\;\;\;\sigma =1,\;2,$$
where $f^{\sigma \tau }$ (the contravariant metric tensor) is obtained as an inversion of the covariant metric tensor $f_{\sigma \tau }=f_{\sigma }\cdot f_{\tau }$ [9] (pp. 295–296), [12]. The basis vectors are defined (see [9], p. 152) $f_{\sigma }=\sum_{\alpha }S_{\sigma \alpha }e^{\alpha }$ and, in our case, have the components: $f_{1}=\left(1;1;0\right),\;f_{2}=\left(0;1;1\right)$. Thus,
(13)$$\|f^{\sigma \tau }\|=\left[\begin{array}{cc} \;\;\;2/3 & -1/3\\ -1/3 & \;\;\;2/3 \end{array}\right]$$

Now it is not difficult to see that
(14a)$$B^{1}=\frac{2}{3}\mu_{X}+\frac{1}{3}\mu_{\mathrm{BX}}-\frac{1}{3}\mu_{B},$$
(14b)$$B^{2}=-\frac{1}{3}\mu_{X}+\frac{1}{3}\mu_{\mathrm{BX}}+\frac{2}{3}\mu_{B}.$$

Combining (11) and (14), we obtain
(15)$$\mu_{X}=\frac{1}{3}A+B^{1},\;\;\mu_{\mathrm{BX}}=-\frac{1}{3}A+B^{1}+B^{2},\;\;\mu_{B}=\frac{1}{3}A+B^{2}.$$

Introducing (15) into (10), we obtain the reaction rate as a proper function of affinities:(16)$$J=(k_{010}\exp\frac{-\mu_{\mathrm{BX}}^{o}}{RT}\exp\frac{B^{1}+B^{2}}{RT}\exp\frac{-A}{3RT}+k_{110}\exp\frac{-\mu_{X}^{o}-\mu_{\mathrm{BX}}^{o}}{RT}\exp\frac{2B^{1}+B^{2}}{RT})(1-\exp\frac{A}{RT})$$

Note that in equilibrium, where A=0, the rate is zero, as expected.

Entropy inequality, or the second law of thermodynamics, puts a restriction on reaction rates expressed as functions of affinities [9] (p. 211). In our case, this restrictive condition is
(17)$${\left(\frac{\partial J}{\partial A}\right)}_{\mathrm{eq}}{\left(A\right)}^{2}\leq 0.$$

Evidently, (17) is fulfilled when ${\left(\partial J/\partial A\right)}_{\mathrm{eq}}\leq 0$. Elaborating on this condition enables to derive additional restrictions on rate coefficients. Carrying out the partial derivative, we come to the following restriction:(18)$$k_{010}+k_{110}\exp\frac{-\mu_{X}^{o}}{RT}\exp\frac{2B_{\mathrm{eq}}^{1}+B_{\mathrm{eq}}^{2}}{RT}\geq 0.$$

This condition can be further modified substituting from (14), (8), and (4):(19)$$k_{010}+k_{110}\left(\exp\frac{\mu_{B}^{o}}{RT}\right)\frac{c_{\mathrm{BX},\mathrm{eq}}}{c_{X,\mathrm{eq}}}\geq 0.$$

Inequality (19) should be valid for any equilibrium (any equilibrium concentrations at a given temperature). Using previously published theorem [15], we conclude that $k_{010}\geq 0$ and $k_{110}\geq 0$. The non-negativity of rate coefficients is consistent with (6) and (7) and the traditional kinetic view on positive rate constants (rate coefficients). This kinetic tradition is followed here, naturally, as a condition to fulfill the entropy inequality (the second law of thermodynamics).

### 2.2. Lotka’s Scheme

This scheme, mentioned in the Introduction, comprises three steps. Houston states [3], p. 70: “Although Lotka mechanism does not (…) correspond to any observed chemical system, its simple mechanism illustrates the basic principles in the more complex oscillatory system.” Here, it serves a very similar purpose—to illustrate the performance of the thermodynamic approach in the case of problematic reaction schemes. In terms of atom-conserving components, Lotka’s scheme could be written as
(R4a)AX+X= 2X+A,
(R4b)X+Y+BY=2Y+ BX,
(R4c)Y=Z.

These steps retain the principal features of Lotka’s steps. Step (R4.1) represents an autocatalytic step for X, whereas (R4.2) represents an autocatalytic step for Y (together with the consumption of X); (R4.3) is the consumption of Y and is identical with the third Lotka’s step. This reacting mixture is composed of four atoms, A=1, X=2, Y=3, B=4, forming seven (n=7) components, AX=1, X=2, A=3, Y=4, BY=5, BX=6, Z=7; the compositional matrix is
(20)$$\|S\|=\left[\begin{array}{ccccccc} 1 & 0 & 1 & 0 & 0 & 0 & 0\\ 1 & 1 & 0 & 0 & 0 & 1 & 0\\ 0 & 0 & 0 & 1 & 1 & 0 & 1\\ 0 & 0 & 0 & 0 & 1 & 1 & 0 \end{array}\right].$$

Its rank (h) is equal to four. Consequently, there are n−h=3 independent reactions in this mixture. They can be selected as follows:(R5a)AX= X+A,
(R5b)X+BY=Y+ BX,
(R5c)Y=Z.

The corresponding stoichiometric matrix
(21)$$\|P\|=\left[\begin{array}{rrrrrrr} -1 & 1 & 1 & 0 & 0 & 0 & 0\\ 0 & -1 & 0 & 1 & -1 & 1 & 0\\ 0 & 0 & 0 & -1 & 0 & 0 & 1 \end{array}\right]$$
fulfills the condition $\left\|P\|{\left\|S\right\|}^{T}=\|0\right\|$, as can easily be checked. As in the previous example, no component can be on both sides of the stoichiometric equation of independent reactions at the same time.

To have autocatalytic steps in the resulting rate equation (thermodynamic polynomial), a third-degree-approximating polynomial should again be used in this case. In this example, the general rate function is a vectorial function [9] (pp. 153-154), whose components are the rates of the individual independent reactions: $J=\left(J_{1},J_{2},J_{3}\right)$. Both the initial and final (simplified) polynomials are lengthy, and more details on their derivation and full forms are given in Appendix A. Here, we reproduce the reduced version of the final thermodynamic polynomial (the rate equation), which retains only the terms corresponding to the independent reactions, or the steps of Lotka’s scheme in the form of classical mass action kinetics (superscripts at concentrations or equilibrium constants mean powers):(22)$$\begin{array}{c} J=k_{1}\left(c_{\mathrm{AX}}-K_{1}^{-1}c_{X}c_{A}\right)+k_{2}\left(c_{X}c_{\mathrm{BY}}-K_{2}^{-1}c_{Y}c_{\mathrm{BX}}\right)+k_{3}\left(c_{Y}-K_{3}^{-1}c_{Z}\right)+k_{4}\left(c_{\mathrm{AX}}c_{X}-K_{1}^{-1}c_{X}^{2}c_{A}\right)\\ +k_{5}\left(c_{X}c_{Y}c_{\mathrm{BY}}-K_{2}^{-1}c_{Y}^{2}c_{\mathrm{BX}}\right) \end{array}$$

Note that the vectors $k_{i}$ contain the rate coefficients (constants) corresponding to p (=1, 2, 3) individual independent reactions (indicated in superscripts); for example, $k_{1}=\left(k_{1}^{1},\;k_{1}^{2},\;k_{1}^{3}\right)$; $K_{p}$ refers to their equilibrium constants. The presence of the autocatalytic terms in (22) is evident, though autocatalytic steps are not among the independent reactions.

The transformation of the rate Equation (22) to the function of chemical potentials is:(23)$$\begin{array}{l} J=k_{1}\exp\frac{\mu_{\mathrm{AX}}-\mu_{\mathrm{AX}}^{o}}{RT}\left(1-\exp\frac{-\mu_{\mathrm{AX}}+\mu_{X}+\mu_{A}}{RT}\right)+k_{2}\exp\frac{\mu_{X}+\mu_{\mathrm{BY}}-\mu_{X}^{o}-\mu_{\mathrm{BY}}^{o}}{RT}\left(1-\exp\frac{\mu_{Y}+\mu_{\mathrm{BX}}-\mu_{X}-\mu_{\mathrm{BY}}}{RT}\right)\\ \;\;\;\;+k_{3}\exp\frac{\mu_{Y}-\mu_{Y}^{o}}{RT}\left(1-\exp\frac{\mu_{Z}-\mu_{Y}}{RT}\right)+k_{4}\exp\frac{\mu_{\mathrm{AX}}+\mu_{X}-\mu_{\mathrm{AX}}^{o}-\mu_{X}^{o}}{RT}\left(1-\exp\frac{-\mu_{\mathrm{AX}}+\mu_{X}+\mu_{A}}{RT}\right)\\ \;\;\;\;+k_{5}\exp\frac{\mu_{X}+\mu_{Y}+\mu_{\mathrm{BY}}-\mu_{X}^{o}-\mu_{\mathrm{BY}}^{o}-\mu_{Y}^{o}}{RT}\left(1-\exp\frac{-\mu_{X}+\mu_{Y}-\mu_{\mathrm{BY}}+\mu_{\mathrm{BX}}}{RT}\right) \end{array}$$

In this example, there are three chemical affinities of the independent reactions ($A^{p},\;p=1,\;2,\;3$) and four constitutional affinities ($B^{\sigma },\;\sigma =1,\;2,\;3,\;4$). Their links to the chemical potentials are shown in Appendix A, together with the full expression for the rate as a function of affinities. Here, the entropic inequality condition, cf. also (17), is a quadratic form in chemical affinities [9] (p. 211):(24)$$-\overset{3}{\underset{p=1}{\sum }}\overset{3}{\underset{r=1}{\sum }}{\left(\frac{\partial J_{r}}{\partial A^{p}}\right)}_{\mathrm{eq}}A^{p}A^{r}\geq 0.$$

Thus, this quadratic form is positive semidefinite. The well-known (Sylvester) theorem of linear algebra states that in this case, all major sub-determinants of the matrix belonging to this quadratic form are non-negative. The matrix can be found in Appendix A; here, we use only the first sub-determinant and the condition corresponding to it:(25)$$-{\left(\frac{\partial J_{1}}{\partial A^{1}}\right)}_{\mathrm{eq}}\geq 0.$$

From it, we have finally:(26)$$k_{1}^{1}+k_{4}^{1}c_{X,\mathrm{eq}}\geq 0.$$

Using the same theorem as above, published in [15], it follows that $k_{1}^{1}\geq 0$ and $k_{4}^{1}\geq 0$. Stating that the numbering of (independent) reactions makes no difference, we can derive, after proper renumbering, additional results: $k_{2}^{2}\geq 0$, $k_{5}^{2}\geq 0$, $k_{3}^{3}\geq 0$.

Now we return to the rate Equation (22) and write explicitly the equations for the rates of the independent reactions, following from it but retaining only those terms that correspond to the tradition of mass action kinetics (i.e., selecting the remaining $k_{i}^{p}$ equal to zero):(27a)$$J_{1}=k_{1}^{1}\left(c_{\mathrm{AX}}-K_{1}^{-1}c_{X}c_{A}\right)+k_{4}^{1}\left(c_{\mathrm{AX}}c_{X}-K_{1}^{-1}c_{X}^{2}c_{A}\right),$$
(27b)$$J_{2}=k_{2}^{2}\left(c_{X}c_{\mathrm{BY}}-K_{2}^{-1}c_{Y}c_{\mathrm{BX}}\right)+k_{5}^{2}\left(c_{X}c_{Y}c_{\mathrm{BY}}-K_{2}^{-1}c_{Y}^{2}c_{\mathrm{BX}}\right),$$
(27c)$$J_{3}=k_{3}^{3}\left(c_{Y}-K_{3}^{-1}c_{Z}\right).$$

The component rates are given, according to the stoichiometric matrix (21), by relations: $J^{\mathrm{AX}}=-J_{1}$, $J^{X}=J_{1}-J_{2}$, $J^{A}=J_{1}$, $J^{Y}=J_{2}-J_{3}$, $J^{\mathrm{BY}}=-J_{2}$, $J^{\mathrm{BX}}=J_{2}$, $J^{Z}=J_{3}$. Selecting $k_{1}^{1}=k_{2}^{2}=0,$ we obtain, from (27), traditional rate equations corresponding exactly to Lotka’s scheme (R4):(28)$$J_{1}=k_{1}\left(c_{\mathrm{AX}}c_{X}-K_{1}^{-1}c_{X}^{2}c_{A}\right),\;\;J_{2}=k_{2}\left(c_{X}c_{Y}c_{\mathrm{BY}}-K_{2}^{-1}c_{Y}^{2}c_{\mathrm{BX}}\right),\;\;J_{3}=k_{3}\left(c_{Y}-K_{3}^{-1}c_{Z}\right)$$
where $k_{1}=k_{4}^{1}$, $k_{2}=k_{5}^{2}$, and $k_{3}=k_{3}^{3}$. Rate Equations (27) and (28) are all consistent with the restrictions on their rate constants derived above on the basis of the entropic inequality. Thus, traditional phenomenological mass-action kinetics is again shown to be consistent with the non-equilibrium thermodynamics theory of reaction kinetics.

## 3. Methods

The methodology has already been described in detail elsewhere. It originated in the paper [16]; its general framework and evolution within non-equilibrium thermodynamics can be found in the book [9]. Specific examples and illustrations of the whole procedure are accessible, for example, in the papers [10,14,15].

The main steps are summarized here:determining the components of a reaction mixture and their atomic composition,finding the number of independent reactions, selecting them appropriately, and finding the corresponding stoichiometric matrix using the linear algebra approach devised and justified by Bowen [12],selecting the degree of the thermodynamic polynomial and writing down the full polynomial,expressing some concentrations from the equilibrium constants of the selected independent reactions and making corresponding substitutions in the equilibrium form of the thermodynamic polynomial,finding restrictions on the polynomial (rate) coefficients, which are followed from the requirement of the general validity of equilibrium [9],introducing these restrictions into the thermodynamic polynomial, thus giving its final, simplified form—the rate equation.
The full polynomial approximates the rate function J=J(T,c) derived by non-equilibrium thermodynamics of linear fluids [9] (p. 248). J is the vector whose components are the rates of independent reactions, and c is the vector of concentrations. The choice of the polynomial degree is guided by the correspondence between the powers of the polynomial terms and the reaction orders; third-degree, at most, should be appropriate; in many cases, first or second degree is sufficient [9] (p. 249). The full polynomial is as follows:(29)$$J=\overset{Z}{\underset{\beta =1}{\sum }}k_{\nu_{\beta }}\overset{n}{\underset{\alpha =1}{\prod }}c_{\alpha }^{\nu_{\beta \alpha }},\overset{n}{\underset{\alpha =1}{\sum }}\nu_{\beta \alpha }\leq M.$$

Here, $c_{\alpha }$ is the molar concentration of component α, and n is the total number of components. The vector $k_{\nu_{\beta }}$ contains polynomial coefficients dependent on temperature only, the vector $\nu_{\beta }=\left(\nu_{\beta 1},\nu_{\beta 2},\ldots ,\nu_{\beta n}\right)$ contains polynomial powers and is also used as subscripts to index various vectors of polynomial coefficients. For the total number of terms Z, see [10,13].

## 4. Conclusions

The application of the stoichiometry-based methods in the chemical kinetics of autocatalytic reactions is rather problematic because these reactions are in certain contrast with the linear algebra of stoichiometry. Construction of rate equations in the form of a thermodynamic polynomial, a technique based on the results of non-equilibrium thermodynamics of linear fluids, has shown how this restriction can be overcome, with autocatalytic step(s) of the rate equations formulated properly. This technique has also enabled the mathematically consistent transformation of the rate equation from the concentration-based form to the form, based either on the chemical potentials or the affinities. Finally, the conditions for the consistency of the derived rate equations with entropy inequality (second law of thermodynamics) have been derived—non-negativity of the rate coefficients (rate constants). The theory illustrated here can be viewed as a tool for verifying and generalizing traditional mass-action kinetics by means of modern non-equilibrium thermodynamics, which is able to deal also with such rather problematic cases.

## Data Availability

No new data were created or analyzed in this theoretical study. Data sharing is not applicable to this article.

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
