# Peer review of "Non-Equilibrium Thermodynamics View on Kinetics of Autocatalytic Reactions—Two Illustrative Examples"

_molecules, 2021, doi:10.3390/molecules26030585_

Round 1

Reviewer 2 Report

The author used a non-equilibrium thermodynamics-based theory of chemical kinetics  to introduce autocatalytic step into conservative rate equations. Further, rate equations based on chemical potentials or affinities were derived and conditions for the consistency of rate equations with the entropic inequality (the second law of thermodynamics) were illustrated.

The whole work is interesting.

POINTS FOR IMPROVEMENT:

  1. A GOOGLE SCHOLAR literature review with the keywords provided by the author revealed about 600 articles published after 2017. Please, complete the literature or/and the provided keywords.
  2. In my opinion the novelty of this work could be highlighted by reviewing the state of the art in the area.
  3. There are no simulations shown. In my opinion the author could add some representative figures and a proper discussion/analysis should be made.
  4. Please connect the illustrative reactions of this work with real autocatalytic reactions.
  5. A very interesting topic in autocatalytic reactions is the existence of complex phenomena as chiral breaking, chaotic attractors or other bifurcations (please see the books of A) R. P. Rastogi "INTRODUCTION TO NON LINEAR PHYSICAL CHEMISTRY and B) Prigogine & Kondepudi) Could the proposed methods of this work applied to chiral breaking, to chaotic systems or to other complex bifurcations? Please, make a proper discussion and/or add examples with simulations

In my opinion this work could be published after revision.

Round 2

Reviewer 1 Report

The author has considered all my remarks, thanks. The rebuttals are clear and the changes done have improved the paper. I can recommend publication in the present form.

Reviewer 2 Report

I am happy with the author's response.